# SCREENED: A Multistage Model of Thyroid Gland Function for Screening Endocrine-Disrupting Chemicals in a Biologically Sex-Specific Manner

**DOI:** 10.3390/ijms21103648

**Published:** 2020-05-21

**Authors:** Lorenzo Moroni, Fulvio Barbaro, Florian Caiment, Orla Coleman, Sabine Costagliola, Giusy Di Conza, Lisa Elviri, Stefan Giselbrecht, Christian Krause, Carlos Mota, Marta Nazzari, Stephen R. Pennington, Annette Ringwald, Monica Sandri, Simon Thomas, James Waddington, Roberto Toni

**Affiliations:** 1MERLN Institute for Technology-Inspired Regenerative Medicine, Department of Complex Tissue Regeneration, Maastricht University, 6229ET Maastricht, The Netherlands; c.mota@maastrichtuniversity.nl; 2Department of Medicine and Surgery—DIMEC, Unit of Biomedical, Biotechnological and Translational Sciences (S.BI.BI.T.), Laboratory of Regenerative Morphology and Bioartificial Structures (Re.Mo.Bio.S. Lab.), University of Parma, 43121 Parma, Italy; fulvio.barbaro@unipr.it (F.B.); giusy.diconza@unipr.it (G.D.C.); roberto.toni@unipr.it (R.T.); 3Toxicogenomics, Maastricht University, 6229ET Maastricht, The Netherlands; florian.caiment@maastrichtuniversity.nl (F.C.); m.nazzari@maastrichtuniversity.nl (M.N.); 4Atturos Ltd., c/o Conway Research Institute, University College Dublin, Dublin 4, Ireland; orla.coleman@atturos.com (O.C.); stephen.pennington@ucd.ie (S.R.P.); 5Institute of Interdisciplinary Research in Molecular Human Biology (IRIBHM), Université Libre de Bruxelles, 1050 Brussels, Belgium; Sabine.Costagliola@ulb.be; 6Food and Drug Department, University of Parma, 43121 Parma, Italy; lisa.elviri@unipr.it; 7MERLN Institute for Technology-Inspired Regenerative Medicine, Department of Instruct Biomaterials Engineering, Maastricht University, 6229ET Maastricht, The Netherlands; s.giselbrecht@maastrichtuniversity.nl; 8PreSens Precision Sensing GmbH, 93053 Regensburg, Germany; Christian.Krause@presens.de; 9UCD Conway Institute, School of Medicine, University College Dublin, Belfield, Dublin 4, Ireland; james.waddington@ucd.ie; 10ARTTIC, 58A rue du Dessous des Berges, 75013 Paris, France; ringwald@arttic.eu; 11Institute of Science and Technology for Ceramics, National Research Council of Italy (ISTEC-CNR), 48018 Faenza, Italy; monica.sandri@istec.cnr.it; 12Cyprotex Discovery Ltd., No. 24 Mereside, Alderley Park, Macclesfield, Cheshire SK10 4TG, UK; s.thomas@cyprotex.com; 13Division of Endocrinology, Diabetes, and Metabolism, Tufts Medical Center - Tufts University School of Medicine, Boston, MA 02111, USA

**Keywords:** endocrine disruptors, in vitro models, bioprinting, omics, decellularization, organoids

## Abstract

Endocrine disruptors (EDs) are chemicals that contribute to health problems by interfering with the physiological production and target effects of hormones, with proven impacts on a number of endocrine systems including the thyroid gland. Exposure to EDs has also been associated with impairment of the reproductive system and incidence in occurrence of obesity, type 2 diabetes, and cardiovascular diseases during ageing. SCREENED aims at developing in vitro assays based on rodent and human thyroid cells organized in three different three-dimensional (3D) constructs. Due to different levels of anatomical complexity, each of these constructs has the potential to increasingly mimic the structure and function of the native thyroid gland, ultimately achieving relevant features of its 3D organization including: (1) a 3D organoid based on stem cell-derived thyrocytes, (2) a 3D organoid based on a decellularized thyroid lobe stromal matrix repopulated with stem cell-derived thyrocytes, and (3) a bioprinted organoid based on stem cell-derived thyrocytes able to mimic the spatial and geometrical features of a native thyroid gland. These 3D constructs will be hosted in a modular microbioreactor equipped with innovative sensing technology and enabling precise control of cell culture conditions. New superparamagnetic biocompatible and biomimetic particles will be used to produce “magnetic cells” to support precise spatiotemporal homing of the cells in the 3D decellularized and bioprinted constructs. Finally, these 3D constructs will be used to screen the effect of EDs on the thyroid function in a unique biological sex-specific manner. Their performance will be assessed individually, in comparison with each other, and against in vivo studies. The resulting 3D assays are expected to yield responses to low doses of different EDs, with sensitivity and specificity higher than that of classical 2D in vitro assays and animal models. Supporting the “Adverse Outcome Pathway” concept, proteogenomic analysis and biological computational modelling of the underlying mode of action of the tested EDs will be pursued to gain a mechanistic understanding of the chain of events from exposure to adverse toxic effects on thyroid function. For future uptake, SCREENED will engage discussion with relevant stakeholder groups, including regulatory bodies and industry, to ensure that the assays will fit with purposes of ED safety assessment. In this project review, we will briefly discuss the current state of the art in cellular assays of EDs and how our project aims at further advancing the field of cellular assays for EDs interfering with the thyroid gland.

## 1. Introduction

The term “endocrine disruptors” (EDs) is used to describe substances that exhibit an “endocrine mode of action” and, thus, may adversely interfere with the activity of the endocrine system. Chemicals that are suspected to be EDs are used for the production of materials commonly present in everyday life. They can be found in food can linings (e.g., bisphenol A); plastics and cosmetics (e.g., phthalates); flame retardants and pesticides (e.g., atrazine and DDT); electric, hydraulic, refrigerant and home appliances (e.g., polychlorinated biphenyls); and even in drinking water and environmental air, primarily in areas around industrial complexes and natural heat sources like geysers and volcanoes (polycyclic aromatic hydrocarbons, heavy metals), to give just a few examples [1].

Since the late 1990s, international regulators, including the European Commission, have started to introduce specific legislations to gradually phase out EDs from the environment. To reach this aim, safety evaluation of chemicals has been developed largely based on 2D cell culture systems, animal testing, and epidemiological clinical studies [2]. In particular, in the case of the thyroid gland function, in vitro tests currently approved by regulatory bodies are based on either 2D cultures of primary thyroid cells or thyroid cell lines, variably suffering from inadequate expression and functionality of basic markers of native thyrocyte activity including thyroperoxidase (TPO), thyroglobulin (Tg) function, or sodium iodide symporter (NIS). Even more, cultures of thyroid gland explant are affected by intrinsic animal-to-animal variability, and tests based on either thyroid stem cells/progenitors or stem cell-derived thyrocytes are lacking. Similarly, animal testing may not offer a thorough picture of changes expected to occur to the human thyroid gland. Finally, though epidemiological studies offer an important statistical tool to understand the size of the problem, the identification of EDs and their effects remains a challenge [3].

In contrast, there is growing evidence that EDs strongly interfere with the mammalian and human thyroid axis at multiple levels, leading to changes in circulating thyroid hormone (TH) concentrations, TH peripheral metabolism, and TH receptor signaling, as well as thyroid gland cytotoxicity [1,4]. However, modes of action (MoA) of EDs on the thyroid axis have just started to be elucidated, and we are still far from having a distinct picture of the different cellular and subcellular pathways involved at any of its anatomical levels (hypothalamus, pituitary, thyroid).

In an endeavor to move away from the ‘black box’ of animal testing towards a toxicological assessment based on human cell responses and a comprehensive mechanistic understanding of cause–effect relationships of chemical adverse effects, the OECD started in 2012 a program to support chemical risk-assessment based on mechanistic reasoning, the so called “Adverse Outcome Pathways” (AOPs). The AOPs are linking key events at different levels of biological organization (responses from cells to organs and then organism) to identify adverse outputs to chemicals. Building on progress in thyroid cell biology, omics technologies, and computational modelling, the complex chains of events that link chemical exposure to toxic outcome in the thyroid gland promises to be unraveled in the near future.

In this project review, we first introduce the limitations of existing tests to assess the effects of EDs on the thyroid gland. Then, we provide a brief overview of the state of the art on tests of EDs on the thyroid axis, in vitro thyroid organogenesis, proteomics and transcriptomics studies of the thyroid tissue, and 3D in vitro assays and recent applications to thyroid cells. Finally, we discuss the feasibility of generating new 3D in vitro models based on an organomorphogenetic approach, which serves as the basis of the new 3D in vitro assays for ED screening that are at the center of SCREENED.

## 2. Limitations of Existing Tests to Assess Effects of Disrupting Chemicals on the Thyroid Gland

A number of in vitro and ex vivo assays are currently available (or under development) to identify and assess thyroid-disrupting chemicals, featuring different characteristics in terms of screening capacity or confirmation of MoA of EDs [3]. Those focusing on TH biosynthesis and possibly integrating multiple MoA at the level of the thyroid gland are particularly attractive to generate robust information on the capacity of a chemical to interfere with the major anatomical structure of the axis, whose direct derangement leads to clinically-relevant conditions spanning from neurodevelopmental disorders to cardiovascular and reproductive system diseases. Although several of the currently available assays are already at a moderately high technology readiness level (TRL ≥ 5, corresponding to OECD levels A and B), they are strongly limited by the availability of adequate amounts of compatible human thyroid tissue and even more by the chances to probe multiple MoA-related AOPs and cytotoxicity of the compounds, primarily when mixtures of EDs are used, and at very low doses [4,5]. Importantly, information on in vivo TH levels and the underlying MoA leading to changes in TH signaling is not available for the majority of these EDs [3]. Indeed, available AOPs are focused on neurodevelopmental effects of EDs [6,7], and no current molecular or cellular assays can clearly distinguish between sex-specific responses of the thyroid gland to EDs. In addition, virtually nothing is known about the effects, MoA and related AOP, of EDs on thyroid stem/progenitor cells, whose role is critical in thyroid development, growth in postnatal life, and adaptation to thyroid injury [3]. Therefore, a robust and reliable 3D in vitro assay able to circumvent these limitations, restrict the variability of the source tissue, selectively identify sex-related responses, and verify the simultaneous action of different EDs, would provide a substantial advancement in the assays for analyzing ED effects on the thyroid function.

Cells in a tissue, organ, and organism are exposed to complex biological environments that are not flat, like classic 2D culture systems, but three-dimensional. Consequently, 2D tissue culture assays far from accurately reproduce the space constraints of cells in vivo. Nevertheless, understanding of biological phenomena is still extensively determined by means of 2D substrates. These surfaces favor a loss of the original cell phenotype (dedifferentiation) during culture, resulting in a different cell behavior that may lead to an abnormal cell activity (e.g., metabolic activity, proliferation, and production of extra-cellular matrix). Furthermore, a complete understanding of the mechanisms driving cellular processes after in vitro culturing on 2D surfaces is still lacking. These problems contribute to the poor correlation between standard in vitro and in vivo observations. In the specific case of the thyroid gland, previous assays that have been used to study thyroid function in vitro include immortal cell lines (e.g., FRTL5, PCCL3, WRT) or primary cultures. Immortalized cell lines are the most frequently used systems, including cell lines derived from human thyroid tumors [8]. Some of them retain the expression of most of the genes involved in TH production (such as NIS, TPO, and Tg), but in general, cells are poorly polarized, which impairs the formation of the functional unit of the thyroid, the 3D follicle. In addition, these cells often lose the receptor for thyroid-stimulating hormone (TSH), a pituitary hormone stimulating thyroid function and growth, thus resulting in a reduced regulation of TH synthesis and release. In thyrocytes, NIS expression, which is responsible for iodide uptake and targeting at the plasma membrane, is modulated by TSH and intracellular organified iodine content. In the absence of follicular structures, iodide organification is impaired. Consequently, an important NIS modulator is lost [9]. In contrast, primary thyroid cells (usually from man, dog, pig, and sheep) are expected to better replicate the mechanisms of thyroid physiology, but their preparation is labor intensive and differentiated features are quickly lost in monolayer. Thus, their use even in short-term cultures is unreliable. In addition, the use of primary thyroid cells from animal origin holds the inherent risk of species–specific differences. Similarly, reorganized primary follicles (in suspension or in collagen) can be maintained in vitro for only a limited period of time [10], and the need for fresh tissue for their preparation constitutes a major drawback, especially with human material; thus, a reliable and new in vitro assay based on differentiated thyrocytes is sorely needed. Finally, due to the limited replication potential of primary thyrocytes, difficulties are encountered for transfections and establishment of stable lines [8].

A mounting body of evidence is now showing that 3D in vitro assays can predict much more accurately than 2D assays and, to some extent in vivo experiments, the effect of chemical compound on cells and tissues. Applications encompass studies of drugs for chemotherapy [11] and renal [12] and hepatic toxicity [13]. Although in thyroid 3D cell cultures knowledge of cell behavior is still limited, recent studies demonstrated that 3D-cultured thyroid follicles are able to respond to morphogenetic signals by modifying NIS expression [14], thus providing a reasonable ground for development of assays based on 3D organization of thyroid cells.

## 3. State of the Art on Screening of ED Effects on the Thyroid Gland

Several studies have shown the effects of different ED compounds on the thyroid gland, including changes in TH production and in activity of key molecules such as TPO, NIS, and Tg. Song et al. demonstrated that environmental chemicals including benzophenones (BPs), polycyclic aromatic hydrocarbons (PAHs), and persistent organic pollutants (POPs) alter TH function by dysregulating TPO activity [15]. PAHs and heavy metals (HMs) have also an effect on thyroid cell cycle by either promoting proliferation (in case of benzanthracene, benzopyrene, and benzofluranthene) or death (in case of cadmium, copper, nickel, and zinc) of thyroid cancer cells [16]. Different phthalates, such as diisodecyl phthalate, dioctyl phthalate, benzyl butyl phthalate, diisononyl phthalate, bis (2-ethylhexyl) phthalate, and dibutyl phthalate also showed the modulation of NIS-mediated iodide uptake, inhibition of cAMP secretion, and a cytotoxic effect [17], most likely due to ROS production, oxidative stress, and elevated levels of thyrotropin-releasing hormone receptor [18]. Polychlorinated biphenyls (PCBs) and methylmercury have also shown toxic effect on thyroid cells, in particular by severely damaging thyroidal structure, decreasing the concentration of TH and inhibiting the gene expression of NIS and Tg through the Akt/FoxO3a/NIS pathway [19,20]. These in vitro studies have been typically performed with cell lines cultured in 2D systems, which are far from replicating the 3D complexity of the actual thyroid gland. In these studies, concentrations of EDs that result in the dysregulation of thyroid cell function range from tens of nM up to hundreds of µM are not always representative of the amounts present in our daily environment. Furthermore, studies investigating combinations of EDs are very limited, if non-existent. Table 1 and Table 2 summarize some of the most relevant observations on the topic.

## 4. State of the Art on Mammalian Thyroid Organogenesis and Recent In Vitro Thyroid Culture Achievements

In mammals and man, the thyroid gland is an endocrine organ located in the neck. It lies anterior and inferior to the thyroid cartilage, where it wraps the first rings of trachea. It is composed of two lobes, joined by an isthmus. In the human, each of the lobes are 4 to 7 cm in length and have a volume of 10 to 15 mL in females and 12 to 18 mL in males [25]. The functional units of this gland are the thyroid follicles, which are spherical structures consisting of thyroid follicular cells or thyrocytes that synthetize the thyroid hormones (THs) thyroxine (T4) and triidothyronine (T3) by iodination of Tg tyrosine residues. THs are important up-regulators of many pathways in the whole organism, ranging from the increase of heart rate, respiration, and gastrointestinal motility to that of carbohydrate and fat metabolism. In the infant, a normal thyroid function is fundamental for brain development [26]. Based on the Human Protein Atlas (https://www.proteinatlas.org/) the three proteins with the highest tissue specificity score in the thyroid gland are Tg, thyroid-stimulating hormone receptor (TSHR), and TPO. This data is in agreement with the predominant function of this gland, which is the production of T4 and T3 hormones (although other cell types such as parafollicular C cells are involved in calcium balance). Thyrocytes have a very slow turnover and divide only a few times during a lifetime. However, the thyroid’s ability to self-renew has been demonstrated. This capacity led scientists to postulate the existence of stem/progenitor cells within the mature thyroid that are able to replenish the pool of fully differentiated thyrocytes [27]. These studies used a number of approaches, including clonogenesis tests, immunocytochemistry, reverse transcriptase PCR (RT-PCR), and fluorescence-activated cell sorting (FACS) to convincingly demonstrate the presence and properties of stem cells within the thyroid [28,29]. In particular, the presence of a population of cells expressing the octamer-binding transcriptional factor 4 (Oct-4), a marker of endodermal pluripotency that is down-regulated during stem cell differentiation, was shown [28]. Moreover, the expression of GATA-4 and HNF-4 in cultured cells derived from the adult human thyroid gland has also confirmed the presence of progenitor cells of endodermal origin [30].

In rodents and man, thyroid development is a complex process involving various morphogenetic events. Early after specification of cells in the primitive ventral foregut endoderm, progenitors of thyrocytes depict a distinct molecular signature, characterized by co-expression of four transcription factors: Hhex, NKX2.1, PAX8, and FOXE1 [31]. Thereafter, thyroid budding occurs, followed by migration of the primordium to a position distant from its site of origin and formation of functional thyroid follicles [31] that express thyroid-specific genes (e.g., Tg, TSHR, TPO, NIS) and produce THs. The development of the embryonic gland and its normal migration are dependent on the interplay between several transcription factors whose knowledge is progressively becoming clear, although still little is known on the intrinsic and extrinsic factors regulating genes involved in thyroid development. In these recent years, the Costagliola lab has established a protocol for efficient generation of functional thyroid follicles from mouse embryonic stem cells (mESCs) in vitro [32,33]. They have shown that a transient overexpression of the transcription factors Nkx2.1 and PAX8 is sufficient to direct mESCs differentiation into thyroid follicular cells (TFC) and effectively promotes self-assembly of TFC into 3D follicular structures in vitro. These follicles produced TH, and their transplantation into athyroid mice rescued TH plasma levels, demonstrating the capacity of mESC-derived TFC to provide functional thyroid tissue in recipients. This in vitro assay provides an unprecedented opportunity to decipher the gene regulatory networks controlling thyroid follicle development and could open new opportunities for stem cell technologies in screening for EDs.

## 5. State of the Art on Proteomics for Investigation of the Thyroid Tissue

Proteomics can be defined as the study of proteomes, consisting of the entire set of proteins present in a biological sample, thus allowing identification of their location, abundance, turnover, and post-translational modifications. Mass spectrometry (MS) is a key method for the investigation of the proteomes of cells and tissues by virtue of its ability to support the simultaneous identification and quantification of thousands of proteins in the same biological sample [34]. The use of appropriate bioinformatics tools enables protein identification, expression, characterization, and localization even in complex protein mixtures [35]. In recent years, proteomics has played an increasingly important role in the characterization of stem cells [36]. A deep characterization and comprehension of the proteome of stem cells is fundamental due to their use in a wide range of applications, including drug screening, regenerative medicine, and cancer research [37]. The investigation of the proteome of adult and embryonic/induced stem cells enabled the identification of new biomarkers of stemness [37].

There has been relatively little data published on the proteomic analysis of the thyroid gland. Using the search term “proteomics and endocrine disrupters” on PubMed results in only 57 publications (as of February 2020). Further analysis of the results reveals as few as 7 studies related to primary research focused on human studies [38,39,40,41,42,43,44]. Of the limited number of thyroid proteomic studies, a large proportion is focused on describing the differences between the proteomes of normal vs. neoplastic thyroid tissue. The first, partially exhaustive, proteomics work was performed using shotgun liquid chromatography (LC)/linear ion-trap (LTQ) tandem MS analysis on pooled protein extracts of normal and tumoral thyroid gland, in which the authors detected 432 proteins in normal and 524 proteins in tumor tissue [45]. A more recent report compared normal thyroid tissue with three common tumors of the thyroid gland, reporting 1629 proteins across all samples [46]. A subsequent study used a Q-Exactive Orbitrap to compare two different types of thyroid tumor with normal thyroid tissues and reported the identification of 2560 proteins [47]. Using a combination of Orbitrap and matrix-assisted laser desorption ionization-time of flight (MALDI-TOF), Gawin and colleagues found a total of 4372 proteins in thyroid tissue [48]. The most extensive analysis of normal thyroid tissue to date has been reported by Liu et al. [49]; by combining filter-aided sample preparation (FASP) [50] and the use of TripleTOF mass spectrometry, they identified 5602 proteins. This level of protein identification facilitates reliable use of gene ontology (GO) and pathway mapping through software such as Ingenuity Pathway Analysis (IPA). Taken together, these results represent the most extensive proteomic analysis of the entire thyroid gland made so far and suggest that glucose metabolism and cell cycle progression are key processes for the function of this endocrine organ.

Compared to thyroid tissue, very little proteomic analysis of thyroid cell cultures has been reported. Pietsch et al. performed a proteomic study on protein interactions in the normal human thyroid cell line HTU-5 and two different cancer cell lines FTC-133 and CGTH W-1 under microgravity conditions [51]. More recently, Bauer et al. analyzed FTC-133 human thyroid cancer cells and used proteomics to elucidate the cellular mechanisms triggering the cells to aggregate into spheroids through the detection of 5924 proteins [52]. Extensive fractionation of samples (by peptide chromatography) prior to LC-MS/MS can increase the number of proteins identified from a typical mammalian cell to between 10,000 and 12,000, although this comes at a cost of requiring larger amounts of cellular material and taking much more MS time. In 2019 Wang et al. [53] analyzed proteomic and transcriptomic data from 29 healthy human tissues, including the thyroid gland. This landmark study provided quantitative data on 18,072 genes and 13,640 proteins from across the human body. In previous studies, protein coding genes have been identified as markers for thyroid cells: Tg, TPO, TSHR, sodium/iodide cotransporter (SLC5A5), forkhead box protein E1 (FOXE1), NK2 homebox 1 (NKX2-1), and paired box gene 8 (PAX8) [54]. However, proteomic studies addressing thyroid specific protein markers are lacking. In silico proteolytic digestion of proteins derived from the translation of thyroid specific genes generates a potential set of peptides specific to thyroid tissue. Processed mass spectrometric data from the Wang et al. study [7] was interrogated further to determine the presence of peptides derived from thyroid tissue markers across the 29 tissue samples analyzed. TPO and TSHR peptides were found exclusively in the thyroid tissue data, and NKX2-1 was identified in the lung and thyroid. These results are not surprising and are in line with previously reported transcriptomic data [55]. Interestingly, TG-derived peptides were also identified in the endometrium, whereas previous reports have been restricted to the thyroid. Peptides derived from SLC5A5 were not present in the thyroid; however, they did appear in the stomach and salivary gland. FOXE1 was identified in the thyroid and tonsil and PAX8 in the thyroid, kidney, and fallopian tube. While the conclusions drawn from this analysis are limited and do not account for peptide intensity, peptides derived from actin, a protein present in all cells, was relatively consistent across all 29 tissues. We therefore anticipate that proteomic signatures can be developed to ascertain the resemblance of stem cell-derived constructs to thyroid gland tissue.

Recently, the range of MS-based proteomics strategies has been extended and diversified, including use of data-dependent and data-independent acquisition strategies, rendering feasible a comprehensive and reproducible analysis of cells and tissues. In addition, analysis can also be extended to common post-translational modifications where desirable or necessary. These studies can be undertaken on a scale that permits robust interpretation of perturbations within systems considering the inherent variability of biological systems (i.e., biological and technical replicates can be included in the MS analysis, while retaining sample throughput and protein coverage). To achieve this, an unbiased label-free single shot LC-MS/MS strategy to provide an optimal balance between depth of proteomic coverage and sample throughput can be used. This approach has been applied to effectively analyze proteomic changes in small amounts of biological materials obtained from regions of tissue sections isolated by laser capture microdissection together with transcriptomic analysis and more comprehensively at a proteomic level using cell lines [56,57].

## 6. State of the Art on Transcriptomics and Its Applications to the Study of the Thyroid Tissue

Since the introduction of toxicogenomics more than 15 years ago [58], it has been a microarray-dominant field, which has largely been focused on three major areas: mechanistic studies, production of ‘reference’ databases, and the development of predictive models [59]. With the arrival of high-throughput RNA sequencing (RNA-seq), the comparison between microarrays and RNA-seq has de facto focused on understanding which of these rival platforms prevails in each of these three applications. The standard toxicogenomics approach usually adopts a general process involving identification of differentially expressed genes (DEGs) resulting from toxin exposure, followed by inference of the functions of the perturbed genes, classically based on analyses of pathway and Gene Ontology. Consequently, a better means of quantifying DEGs is a primary concern for correct biological interpretation. The FDA-led sequencing quality control project (SEQC) [60] compared RNA-seq to gene expression microarrays through a comprehensive study design using many chemicals that act through diverse modes of action. The study concluded that although the two platforms performed comparably for the highly expressed genes, RNA-seq had a better sensitivity than microarrays for weakly expressed genes [60]. Currently, it is accepted that the RNA-seq offers an advantage when differences between treated samples and matched controls are expected, and, primarily in the case of studies with toxins at low dose, lead to high improvement in toxicogenomic outcomes. Thus, SCREENED logically endorsed the use of next-generation sequencing (NGS) RNA-seq in its program.

RNA-seq not only offers an improved measurement of DEGs, but also detects other transcriptomic events, some of which are difficult to monitor by means of microarray technologies. RNA-seq is expected to advance the “comprehensiveness” of toxic modes-of-action. An obvious advantage of RNA-seq refers to its capability to access splicing variants [61]. Through NGS, the ENCODE project revealed that close to 95% of human multi-exon genes undergo alternative splicing [62]. If the consensus is that the majority of genes are regulated through their mRNA level, NGS showed that some genes stably expressed are mainly regulated by selection of their splicing variants. On similar evidence, it has been suggested that up to 60% of human diseases could be due to splicing perturbation [63]. Thus, beyond the main gene profiles, the study of splicing perturbations induced by EDs to our thyroid model can be considered vital.

RNA-seq has also emerged as a robust tool for microRNA (miRNA) detection [64]. MiRNAs are a class of regulators of gene expression that have important roles in disease pathogenesis and toxicity [65]. MiRNAs expression levels are very sensitive to changes of phenotype and adaptation to chemical exposure, making them promising biomarkers in many fields of research. Several studies are now considering the investigation of the miRNA level circulating in the blood stream as putative early biomarker for diagnostic of toxic and carcinogenic responses. Indeed, many research groups have studied the miRNA profile of thyroid cancers [66], including the putative implication of EDs in this process [67]; however, little is known of the thyroid-specific response to EDs. SCREENED will then be able to investigate in depth the miRNA profiles of thyroid cells exposed directly to EDs. Finally, NGS allows not only the ability to assess the expression of each miRNA main mature form, but also to review the complete profile of alternative microRNA molecules (called “isomiRs”). Preliminary data on biological systems different from the thyroid gland (not yet published) seems to indicate that a possible response to toxic compounds induces a general shortening of miRNA profiles (and thus a global bias after exposure to shorter miRNA isomiRs).

Currently, the field of thyroid toxicogenomic research is underrepresented. Few studies have been performed analyzing the effects of endocrine disruptors on thyroid models in vivo or in vitro utilizing transcriptomics approaches like microarray or RNA-Seq [68,69,70,71]. An exhaustive search in public repositories for thyroid transcriptomic data, such as ArrayExpress, Gene Expression Omnibus (GEO), and the European Nucleotide Archive (ENA) reveals how the majority of the datasets available derive from normal tissues, primary tumors, or cell lines. However, very few datasets originate from samples exposed to endocrine disruptors. Porreca et al. analyzed the effects on the transcriptome of rat PCCL3 cells to the exposure of ethylenethiourea (ETU) and chlorpyrifos (CPF), two known thyroid-disrupting chemicals [68]. Song et al. tried to predict the effects of TPO disruption on transformed FTC-238 human follicular thyroid carcinoma cells constitutively expressing TPO treated with compounds that either increase or decrease TPO activity [72]. In a study by McDougal and colleagues, the changes in the rat thyroid transcriptome after perchlorate exposure were analyzed [73]. It appears evident how there is currently a data gap concerning the effects of endocrine disruptors on thyroid models. While already scarce, raw data availability and clear metadata description are sometimes insufficient to ensure the interoperability of these public datasets into meta-analysis with other newly generated datasets.

Though multi-omic approaches are very powerful tools, they are not extensively carried out, mainly due to cost and time limitations [74]. They are employed in a variety of different contexts, such as precision medicine [75] and oncology [76,77], disease research [78,79], and physiology [80]. Since the correlation between mRNA and corresponding protein levels is very variable [81,82,83], a look at both aspects is necessary to obtain a comprehensive molecular view.

## 7. State of the Art on In Vitro 3D Culture Assays and Recent Applications to the Thyroid Cells

Up to now, high-throughput screening of the effects of EDs on humans have either been studied in 2D conventional in vitro assays of the respective cells/tissues or by simplified 3D microtissues, which only partly reflect the organization and functionality of the native tissue. The rapidly developing field of organoid technology has shown that organoids are well-advanced model systems which better mimic organ function and architecture in a dish [84]. However, one of the key shortcomings of organoids is the lack of standardization [85]. The high heterogeneity of organoids with regard to their size, shape, cellular architecture/composition, and maturation of cells limits their broad use in high-throughput screening assays. Microengineered wells have widely been used as tools to better control the formation of more homogeneous spheroids and embryoid bodies in the past [86], but have not been explored so far to improve the formation and maintenance of organoids [87].

To identify and address the effect of EDs on the thyroid gland in a more controlled manner, microfluidic bioreactors can be used. Conventional bioreactors, such as stirred tank bioreactors, are widely used for large-scale industrial production of biologics and biopharmaceuticals [88] because these systems provide excellent control over important culture conditions, such as oxygen tension and shear. Miniaturized bioreactors are commonly used in the field of tissue engineering in order to overcome diffusion limitations in 3D cell aggregates exceeding critical dimensions. Bioreactors in the field of organoid cultures have not been used so far, although stem-cell based organoids easily reach similar or even larger sizes, which might limit their long-term culturability [89]. The same debate has been recently begun in the field of 3D bioprinting where bioreactors are also considered as one of the central tools in the future to create and maintain functional bioprinted tissues or organs in vitro [90].

To yield precise spatial–temporal homing of thyroid cells onto the 3D in vitro constructs, magnetically labelled cells can be generated for their control and exact position in the bioreactor core. Nowadays, magnetic nanoparticles are receiving significant attention for a wide range of medical applications. Indeed, they have been progressively employed as support materials for enzyme immobilization, targeted drug or gene delivery, tumor treatment by hyperthermia, contrast agents in magnetic resonance imaging (MRI), to magnetically label cells in cell therapy, and many other exciting biotechnological applications [91,92]. Superparamagnetic iron oxide nanoparticles (SPIONs) are the only clinically approved metal oxide nanoparticles; however, the exposure to SPIONs has always been associated with significant toxic effects [93]. Recently, the Tampieri lab developed an innovative biocompatible and bioresorbable nano-hydroxyapatite (FeHA NPs), where highly controlled Fe^2+^/Fe^3+^ ion doping results in defined occupation of calcium crystal sites in the apatite lattice, providing the final molecule with an inherent superparamagnetic ability that avoids the presence of poorly tolerated iron oxide secondary phases [94,95]. FeHA NPs do not reduce viability of the cells currently tested and enhance cell proliferation and viability compared to a classical hydroxyapatite compound. Indeed, FeHA NPs can be internalized by human mesenchymal stem/stromal cells to obtain magnetically labelled cells that, through the application of an external static magnetic field at low intensity, can be positioned at designed targets in the tissue. Exploiting this property, we expect that magnetically-labeled cells can be guided to specific sites in the fabricated 3D constructs. As a result, a fast and selective seeding would occur, thus boosting the process of eventual scaffold colonization and 3D structure formation. The main advantage of this specific type of magnetic particle with respect to the commercial ones is their high biocompatibility. In fact, FeHA NPs have been shown to be easily incorporated into the recipient cells by endocytosis, without negatively affecting their proliferation, morphology, and differentiated state. Thus, the principle of the remote control of magnetized cells coupled to that of an organ-tailored, biocompatible matrix for their seeding inside a bioreactor is expected to maximize the chances for ex situ assembly of thyroid constructs, predicted to be structurally and functionally consistent with a natural thyroid gland.

A further resource of the biotechnological array that is available to the SCREENED project is bioprinting. Bioprinting is a novel and fast developing group of techniques that allows the manufacture of tissue-engineered in vitro assays by selectively dispensing cells, hydrogels, and combinations of these inside a defined 3D space. These techniques arose from simplified industrial techniques termed “additive layer manufacturing” where polymers and ceramics were used to manufacture 3D objects. These additive manufacturing techniques have been adapted and exploited in the field of tissue engineering for the manufacturing of temporary scaffolds, mainly for skeletal applications, that could provide mechanical support as well as sufficient porosity to allow new tissue ingrowth, access to nutrients, and metabolic waste exchange. Conversely, bioprinting uses hydrogels that normally present much lower mechanical properties, but allow the morphogenesis of new tissue while maturation occurs post-printing, in a way similar to what occurs during morphogenesis in vivo. To permit this “morphogenesis-like” process, a careful selection of hydrogels, cells, and biological factors must be taken into account. Over the last decade, several in vitro assays have been manufactured with bioprinting techniques spanning from epidermal-dermal, hepatic, cardiac, renal, cartilaginous, and many other tissues. Some of these assays, mainly those aimed at recapitulating skin and kidney structures, reached the market and are currently used as toxicity tools for cosmetics and pharmaceutical industries, as reasonable replacements of in vivo animal testing [96,97]. Based on these initial examples, it is clear that such a strategy might soon provide reliable biotechnological platforms to replace in vivo testing on experimental animals; thus, both academia and industry are working closely to develop and launch on the market these in vitro assays also for a number of other human tissues. In the literature, only few examples of in vitro assays based on bioprinted glands are currently available [98,99]. The thyroid gland is one of the examples where researchers used bioprinting to selectively dispense spheroids (thyroid and endothelial) to produce a thyroid-like tissue [98] that, upon implantation, showed some degree of functionality in a hypothyroid mouse model.

Biological tissues are 3D structures and offer a good example for development of 3D assays. When engineered, they are usually grown in various, small devices like microfluidics or multiwell plates. However, in many cases, no control and even no monitoring of the basic culture conditions is available including oxygen tension and pH, although these parameters are essential for the survival and functionality of the cells. Most of the analytical methods currently applied are either based on off-line methods with sampling of cells or media or on destructive methods, like cell and tissue fixation and staining. Such methods are time consuming and not adequate to control the process conditions in real-time, e.g. to adjust oxygen contents in the media or to trigger exchange of media when nutrients are consumed. During quality control of tissue/cell material for pharmaceutical tests, reliable and reproducible results can only be achieved if starting materials are identical or at least comparable. In the case of cells and their 3D constructs, this can be ensured by sampling a defined amount of material and scarifying it for characterization. However, in 3D constructs, spatial inhomogeneity is frequent, thus a non-destructive and ideally non-invasive method to characterize or qualify the tissue to be used in a test or assay is a robust prerequisite for the acceptance in the market. For these reasons, the use of chemical-optical sensors to measure critical culture parameters is advisable. Basically, these sensors are composed of an indicator-dye immobilized in a polymer matrix. They can be applied as a dot inside a transparent vessel to measure in the bulk media or as a coating on a light guide, which can be even inserted into the 3D construct. These sensors use fluorescence lifetime detection to reduce influences of instrumentation and experimental set ups. The fluorescence decay time of the immobilized indicator dye is dependent on the oxygen partial pressure. A pulsed flash of light is sent through an optical light guide to the sensor, and the light sent back from the dye is analyzed inside the instrument using a phase detection method [100]. Oxygen and pH sensors based on this principle were successfully commercialized years ago and can be found in many different applications. Especially in the market of disposable bioreactors, they have ruled out all other technologies. Due to their outstanding robustness and reliability, optical oxygen sensors based on fluorescence decay time also replaced oxygen electrodes in other applications like process control, food and beverages, and others. Optical sensors for real-time monitoring of oxygen, and in rarer cases of pH, were applied in different tissue generation processes including medium-scale perfusion systems and microfluidics [101]. Microsensors were also used to measure oxygen on-line inside the tissue; this method gives a very precise and unique view “inside” the tissue during formation and enables a controlled process to yield uniform and repeatable quality of measurements [102].

## 8. Beyond Current 3D In Vitro Assays: Replicating the Anatomy and Functionality of the Thyroid Gland

In SCREENED, we address the previously highlighted limitations by fabricating a modular microbioreactor able to house three-dimensional (3D) in vitro assays of a thyroid gland based on advanced tissue engineering and biofabrication technologies. These 3D in vitro assays will be used to screen and evaluate comprehensively the effects of different EDs at an unprecedented level of molecular detail thanks to an integrated omics approach. We expect our assays to be more sensitive and therefore able to reflect changes at concentrations relevant to in vivo exposure. We will make use of three different advanced 3D in vitro assays, namely thyroid organoids and engineered thyroid constructs obtained by using either a decellularized thyroid lobe matrix or 3D bioprinting. These constructs will be transferred into the modular microfluidic bioreactor by using the same type of carrier, a thin 3D shaped porous polymer film or membrane. Membranes will mainly be used as joint and standardized substrates for the decellularized and 3D-bioprinted constructs and their insertion into our microfluidic bioreactors. Differently, the microwell arrays from the same material, produced by Substrate Modification And Replication by Thermoforming (SMART) technology [103], will be used to advance the field of organoids for screening purposes. Our microwell arrays will help to reduce organoid heterogeneity by providing microscale confinements with defined sizes and shapes. The microwell arrays will facilitate the formation of organoids at defined positions in space. This will pave the way for screening high numbers of standardized thyroid follicles/organoids by means of fully automated imaging techniques and, possibly, real-time mass spectrometry, helping to save time and costs particularly in industrial settings.

Another major issue we expect to overcome is the maintenance of homogeneity in the cell population during its seeding on 3D porous scaffolds, as in the case of a 3D matrix from a decellularized thyroid lobe or bioprinting of a 3D construct. In SCREENED, in fact, we will take advantage of cell-internalized magnetic particles and the integration of microcoils to generate a defined, spatially patterned magnetic field in our microfluidic device. So far, planar microcoils have been integrated into lab-on-chip devices for analytical applications, e.g., microcoil NMR on chip [104]. Besides NMR, magnetic fields created with planar microcoils in chips have been used to locally control and separate magnetic beads and particles in various microfluidic lab-on-chip applications [105]. Moreover, planar microcoils and microcoil arrays have also been used to control the position of cells in microfluidic compartments or channels [106,107]. In SCREENED, we will develop an improved dynamic cell seeding process (e.g., onto a construct based on decellularized matrices) by the superposition of flow and a dynamically controlled, spatially patterned magnetic field. Cells with internalized magnetic particles will be steered by our patterned magnetic field to coordinates within the 3D scaffolds where otherwise lower cell densities would occur.

Using trophic perfusion at high accuracy, our technology will allow for real-time delivery of different EDs to the thyroid cells. Furthermore, sensing technology will also be integrated for real-time monitoring of the metabolic stability of the growing environment around thyroid cells during their self-assembly as a 3D thyroid organoid and to detect effects of the tested EDs on these vital metabolic parameters.

As recently shown by Toni’s lab [108,109], the central idea in SCREENED is that it is possible to reliably replicate the function of a native endocrine organ by reconstructing its 3D replica with stem/progenitor cells induced to differentiation on a 3D biocompatible scaffold recapitulating the 3D geometry of the organ’s natural matrix (so called organomorphism). As a result, cell surface signals (i.e., topobiological inputs) and mechano-chemical transductions may take place to guide seeded stem cells/progenitors up to their self-organization into a 3D functional organoid first, and a replica of an entire thyroid gland in further developments of our model. Indeed, the structure–function relationship is very strong during development in the thyroid gland [110]. Thus, getting functional insights also at an intermediate level of reconstruction like that offered by 3D organoids may represent a powerful informational supply to conduct the analysis of the EDs’ effect on a final construct, as that relies on an in vitro recellularized 3D stromal matrix of the thyroid lobe or a bioprinted model.

Compared to conventional 2D culture assays, the proposed 3D culture in vitro assays have several advantages (Table 3). Organoids, decellularized extracellular matrix (ECM), and bioprinted constructs all provide significantly enhanced cell-to-cell and cell-to-ECM interactions. These 3D systems allow for co-culture with other cells of the tissue microenvironment including endothelial, neural, muscular, and immune cells. Decellularized ECM retaining the 3D geometry of the native stromal matrix then ensures an even greater recapitulation of the tissue microenvironment and related biological properties, although differences in the microarchitecture and its composition may ensue as a result of different decellularization procedures and sex of the donor. Conversely, bioprinting allows for an exquisite control over the structural architecture but does not yet recapitulate many features of the tissue ECM. Yet, in combination with decellularized ECM, bioprinted constructs have the potential to get close to the organization of the native tissue microenvironment. To our knowledge, no study is currently available in the literature comparing these three different main 3D cell culture systems; for this reason, we will develop three different 3D thyroid in vitro assays based on the previous assumptions, to assess the most effective set of parameters able to provide a thyroid function suitable for toxicological studies in a 3D culture.

## 9. Sex-Specific, Structural and Functional Dimorphism of the Mammalian Thyroid Gland and Strategies to Address this Topic in the Pursued Laboratory Modelling

In mammals and man, a striking sex-specific structural and functional dimorphism characterizes the thyroid gland. Volume, weight, and lobar diameters are proportionally higher in men than women, primarily depending on a higher male body weight [111]; in addition, pre-pubertal and adolescent females have a higher thyroid volume than males up to age 30, when the male volume becomes preponderant [112]. Finally, at all age intervals women have higher FT4 values than men [113] with FT4 peaks in relation to follicular and lutheal phases of the menstrual cycle [114] suggesting an overall higher thyroid hormone biosynthesis.

Similar to humans, male rats presents bigger follicular cells and higher volume of the epithelial fraction with respect to female rats in estrous, although males harbor a lower amount of intrafollicular colloid in comparison to females [115], suggesting a lower thyroid hormone reservoir in the male sex. Consistently, male rats have lower T3 levels than females, whereas thyroid deiodinase type 1 activity is lower in female than in male rats [116]. After puberty, then, the volume of the thyroid epithelial fraction is reduced in females [117], indicating a sex-related control of the mature thyrocyte size.

To fulfil sex-related specificities in thyroid structure and function, our proposed 3D in vitro models of thyroid constructs will include the use of thyroid stromal matrices and cell sources originally coming from male and female donors, to preserve as much as possible even at molecular levels of potential sex-related features and sensitivities. These include the cytoskeletal/cytoplasmic organization and signaling systems featuring peculiar membrane proteins, receptors, and adhesion molecules. Indeed, it is now clear that sexual dimorphism is intrinsic to a number of adult stem cell/progenitors of mesodermal, neuroectodermal, and endodermal lineages. Muscle-committed satellite cells are found in higher number in women than men where they, however, result bigger and with less muscular regenerative potential [118]. Similar, in vivo and in vitro activation, growth factor secretion, regenerative potential, and number of adult stem cells/progenitors from epidermal hair follicles, sebaceous glands, liver, bone marrow, and central nervous system are higher in female than male rodents, as opposed to the kidney and skeletogenic and chondrogenic mesenchymal stromal cells where responses prevail in the male sex [118]. Finally, recent observations suggest that even mESCs from eight-cell embryos express sexually dimorphic genes leading to transcription of sex-linked autosomal genes in the post-implantation blastocyst [119,120,121], whereas in humans, pluripotent stem cells (either ESCs or induced pluripotent stem cells—IPSCs) may depict prominent male features at the time of X inactivation. This suggests that their laboratory reprogramming does not abolish their original sex signature [122]. Collectively, all these data support our assumption that thyroid-specific 3D constructs are an adequate choice to reveal sex-related differences in cellular and molecular responses to a variety of EDs.

## 10. Feasibility of the Organomorphism Approach

ESCs emerged as a promising model to dissect and recapitulate the molecular events and gene networks that regulate cell fate determination, cell differentiation, and organogenesis. Only very recently, Antonica et al. achieved the successful generation of functional thyroid tissue from mESCs, after a transient Dox inducible co-expression of Nkx2.1 and Pax8, two transcription factors involved in thyroid development [33]. The employed differentiation protocol allowed for the efficient generation of thyroid follicular cells, which organized in vitro into 3D follicular structures capable of iodide organification, a hallmark of mature thyroid tissue. When grafted in vivo into thyroid mice, the mESC-derived follicles rescued thyroid hormones plasma levels and promoted subsequent symptomatic recovery. In vivo experiments demonstrated that mouse models of thyroid development and function recapitulate observations made on hypothyroid patients. Therefore, the use of 3D mouse organoids appears a perfect tool for studying effects of EDs on thyroid cell function. However, unforeseeable effects related to subtle species-dependent variations in thyroid physiology cannot be ruled out. Ma et al. generated thyroid cells from human ESCs, which produced cAMP upon TSH stimulation, but T4 production was not reported and a sustained induction of ectopic NKX2.1 and PAX8 was necessary to maintain thyroid cell differentiation [123]. Using a different approach, based on manipulation of bone morphogenetic protein (BMP) and fibroblast growth factor (FGF) pathways, Kurmann et al. induced thyroid fate in mouse and human stem cell derived-endodermal precursors. This protocol opens new perspectives in the generation of human thyroid progenitors. Nevertheless, the protocol needs improved efficiency and lengthening of cell survival in culture [124].

In their seminal work, Collins et al. [125] proposed the bypassing of animal-based human safety testing by shifting toxicology to a predominantly predictive science focused on broad inclusion of target-specific, mechanism-based, biological observations in vitro. In SCREENED, a key is to develop bioactivity profiles of EDs that are predictive of deranged thyroid function, through identification of signaling pathways, which lead to toxicities upon perturbation. This mechanistic information is then to be used to iteratively develop computational models that can simulate the kinetics and dynamics of toxic perturbations in pivotal intracellular transduction chain of events, ultimately leading to systems models that can be applied as in silico predictors for human drug safety. These models will then be framed into AOPs progressively built up with functional and omics-based molecular data generated in vitro from our innovative 3D thyroid assays.

In preliminary experiments using protein extracts from thyroid cells subjected to digestion using the FASP protocol and analyzed on a Thermo Fusion LC-MS/MS, we have been able to identify close to 5000 proteins in a single shot analysis. We will use this approach to undertake proteomic analysis of the three different 3D in vitro thyroid assays and to investigate the impact on them of EDs. The comprehensive proteomic data will be analyzed extensively, using a range of pathway, GO, and protein network analysis approaches. In this way, we will be able to i) confirm the authenticity of the 3D in vitro assays and ii) identify signatures of proteins that support the characterization of EDs. These signatures may then be measured and monitored by targeted proteomic strategies with multiple reaction monitoring assays applied to panels of proteins that are generated and the proteins then measured robustly, using triple quadrupole mass spectrometers. With the availability of transcriptomic data, we will also be able to undertake integrated proteogenomic analysis using RNA sequence data to generate databases with which to search the peptide MS data. The development of multiplexed panels of proteins makes sense by virtue of the fact that in response to treatments—including EDs—multiple biological networks become perturbed in the cells, and hence, changes in relevant proteins from these networks can be chosen as parameters for evaluation of the assays’ outputs.

Development of multiplexed immunoassay tests is very challenging because of significant technological limitations, including availability of reagents, interference, cross-reactivity, and lack of specificity [126]. In particular, cross-reactivity is an enormous challenge, especially in complex mixtures of analytes, such as biological tissues. In contrast, multi reaction monitoring (MRM)-MS is a technology platform that can overcome many of these challenges, primarily due to its high specificity, high multiplexing capabilities, and low cost of assay development [127]. We will exploit the advantages of MRM-MS to develop assays for measuring ED responses in the three different 3D in vitro thyroid constructs. In SCREENED, we will consider the transcriptomics investigation to the level of transcript isoforms, and thus be ultimately able to propose putative toxicity biomarkers involving a switch in the transcript usage of genes, particularly relevant for the cases where the global gene expression would not be affected by the EDs. Alternative splicing, although being the main cause of gene variation, is not the only source of isoforms. Alternative polyadenylation, RNA editing, gene fusion, and alternative transcription start site all contribute to the mechanisms capable of producing transcripts different from the canonical form. All these variants can potentially redefine the known molecular metabolic pathways and will then be considered in our analysis from data generated by transcriptomics. As a result, we will investigate this possible occurrence after ED exposure, and thus potentially propose a global miRNA isomiRs profile perturbation as a putative biomarker of ED toxicity.

## 11. Conclusions

SCREENED will move beyond the current state of the art of in vitro assays evaluating effects of EDs on the thyroid function. To reach this, we will develop three different and innovative 3D in vitro assays, expected to shed light also on effects involving thyroid stem/progenitor cells. The three different 3D in vitro assays are based on: 1) a 3D organoid based on stem cell-derived thyrocytes, 2) a 3D organoid based on a decellularized thyroid lobe stromal matrix repopulated with stem cell-derived thyrocytes, and 3) a bioprinted organoid based on stem cell-derived thyrocytes able to mimic the spatial and geometrical features of a native thyroid gland. 3D dynamic cell culture will be enabled by a new modular microbioreactor monitoring metabolic and functional parameters of the thyroid biology. This will allow for exposure to low doses of EDs, and control of the geometrical placement of cells through magnetic fields. As a result, we expect to increase the structural homogeneity of the 3D thyroid constructs and achieve an anatomical mimicry of the native thyroid gland. The results obtained in SCREENED from the transcriptomics analyses will contribute to creating a comprehensive description of thyroid transcriptome alterations following exposure to selected EDs and will be integrated with proteomics data using both a targeted and untargeted approach. Adopting an untargeted approach, RNA expression and protein levels from the same sample can be combined to perform a proteogenomic characterization of ED exposure, to describe the cellular response at the molecular level. In a targeted approach, it will be possible to select gene/protein signatures indicative of ED exposure that will be used to develop targeted proteomics assays. The final objective is to select a set of targets that can be used as molecular biomarkers of individual ED exposure. Due to the lack of both proteomic and transcriptomic data on thyroid tissue, the SCREENED consortium is also currently performing a pilot study for the identification of both omics in untreated vs. ED-treated thyroid-derived cells. ‘Normal’ cell lines are to be preferred to cancer cell lines for toxicogenomics studies, since they retain some key thyroid functions and gene expression [128,129]. This pilot study is anticipated to provide a baseline measurement to inform the research planned for the stem cell-derived constructs, where limitations in cellular material exists. In both omics platforms, discovery experiments will be used to identify changes in molecular signatures and infer molecular pathway disruption. These changes can be incorporated into targeted assays, to improve both assay sensitivity and quantification capabilities, which in turn requires smaller amounts of cellular material. This will be combined with a robust in vitro/in vivo experimental extrapolation by proteogenomic analysis and biological modelling, enabling the consortium to determine new MoA and related AOPs for a number of EDs.

## Figures and Tables

**Table 1 ijms-21-03648-t001:** Doses of EDs normalized to the same unit of measurement and their effects on human thyroid cell monolayers.

		Human Thyroid Carcinoma FTC-238 Cells Male Donor [15]	8505C Human Thyroid Cells Female Donor [16]	KAT5 Human Thyroid Carcinoma Cells Female Donor [21]	TT Human Thyroid Cell Line Female Donor [19]
**Polycyclic aromatic hydrocarbons (PAH)** **µg/L**	Benz(a)anthracene (BAA)		220 **↑ cell proliferation**		
	Benzo[a]pyrene (BAP)	252.31 **↓TPO**	250 **↑ cell proliferation**		
	Benzo[k]fluoranthene (BKF)	0.25–2.52 **↓TPO**	250 **↑ cell proliferation**		
	Dibenzo[a,h]anthracene (DahA)	278.34–2783.46 **↑TPO**			
	3-methylchloranthracene	0.02–214.69 **↓TPO**			
	Pyrene	0.20–20.22 **↓TPO**			
**Heavy Metals** **µg/L**	Cadmium (Cd)		160 **cytotoxic effect**	560–1120 **arrest of cell growth in the G0/G1 phase**	
	Copper (Cu)		90 **cytotoxic effect**		
	Nickel (Ni)		90 **cytotoxic effect**		
	Zinc (Zn)		100 **cytotoxic effect**		
	MeHg				107.81–4312.54 **cytotoxic effect**

**Table 2 ijms-21-03648-t002:** Doses of EDs normalized to the same unit of measurement, and their effects on thyroid cell monolayers from different mammalian species including humans.

		FRTL 5 Rat Thyroid Cells Unspecified Karyotype [17]	Primary Human Thyroid Cells Unspecified Karyotype [22]	FRTL 5 Rat Thyroid Cells Unspecified Karyotype [23]	Human Thyroid Follicular Rpithelial Cell Line (Nthy-ori 3-1) Unspecified Karyotype [18]	TT Human Thyroid Cell Line Female Donor [19]	Human Thyroid Epithelial Cells Female Donor [20]	Primary Pig Thyroid Cells Unspecified Karyotype [24]
**Phthalates** **µg/L**	DIDP	44,600–446,000 **↑ Iodine uptake**						
	DOP	39,000–390,000 **↑ Iodine uptake**						
	DINP	41,800–418,000 **↑ Iodine uptake**						
	DEHP	39,000–390,000 **↑ Iodine uptake**	3900 **↓cAMP**		39,060–156,240 **↑ cell proliferation ** **↑ROS production** only at 156,240 **↑TRH-R**			
	BBP	31,200–312,000 **↑ Iodine uptake** **cytotoxic effect**						
	DBP	80,400–804,000 **cytotoxic effect**						
	MEHP		27,830 **↓cAMP ↓ TG ** **cytotoxic effect**					
**Organophosphate flame retardants (OPFRs)** **µg/L**	Triphenyl phosphate (TPP)			1000 **↓TG ↓TSH-R** 3000–10,000 **↑ NIS** 10,000 **↑ NIS ↑ TPO**				
**Polychlorinated biphenyls (PCB)** **µg/L**	PCB 118						0.81–8.16 **↓ TG ↓T4** 81.60–8160.55 **cytotoxic effect**	
	PCB 126					979.26 **cytotoxic effect**		99.88 **↑ cytochrome P- 450** **↓ NIS**

**Table 3 ijms-21-03648-t003:** Comparison between 2D and 3D cell culture assays.

Parameter	2D Assays	3D Assays		
		Organoids	Decellularized ECM	Bioprinted Constructs
Cell-to-Cell, Cell-to-ECM Interactions	Poor	Excellent	Excellent	Excellent
Tissue microenvironment	Extremely Limited	Limited	Good Duplication	Limited
Control of structural architecture	No	No	Yes, but influenced by chemical and physical preparations	Yes
Presence of connective tissue cells	Possible but not used	Yes	Yes	Yes

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
