# Peer review of "SCREENED: A Multistage Model of Thyroid Gland Function for Screening Endocrine-Disrupting Chemicals in a Biologically Sex-Specific Manner"

_ijms, 2020, doi:10.3390/ijms21103648_

Round 1
Reviewer 1 Report
This is an interested review of SCREENED EU Project under Horizon2020 on an effect of EDs in developed 3D in vitro models of human thyroid gland. This review is of interest, clearly organized and well described, with novelty aspects but it needs several corrections before accepted for publication.
- Please rethink whether you describe thyroid axis or rater thyroid gland? To add, the thyroid endocrine system is controlled by hypothalamus–pituitary–thyroid axis(HPT).
- A biologically sex-specific aspects of thyroid function and in your SCREENED 3D model might me described slightly more comprehensive.
- Please rethink the setup of presented tables. To add, in the Table 1 there are only data from human in vitro studies (see the tables caption). It would be interesting to add the origin of specific cell lines, mainly including sex of donor.
Author Response
Reply: The reviewer likes to have a specification of the site of analysis that in the current model is focused to the effects of EDs on the thyroid gland. We have modified the manuscript accordingly, where originally it was indicated “thyroid axis”, to substitute with “thyroid gland”. Only where it is still was appropriate to leave “thyroid axis” due to a general reference to the three parts of the axis (hypothalamus, pituitary, thyroid) the term can be left as such.
As requested by the reviewer, a short section on sex-specific differences in the thyroid gland structure and function, and predictions how to address this in the thyroid model, is provided.
It is not clear the meaning of “rethink the setup of presented tables”. Tables refer to what currently published in the international literature specifically referred to actions of EDs known to have direct effects on thyroid cells. This point is already stated in the section present in the original text at “State of the art on screening of ED effects on the thyroid axis”.
In addition, Tables include immediate evidence on the effects observed for each EDs tested, and the identical unit of measurement for the doses used facilitates comparison on dose-response results between different compounds. The aim of the tables is to describe direct effects of EDs on thyroid cells avoiding in vivo results that may harbor confounding effects occurring at various levels of the axis (either hypothalamus, and/or pituitary, and/or thyroid). The reason for this choice is that the 3D modelling of the project is related to the thyroid gland and not to the analysis of the entire axis. However, the reviewer notes that Table 1 only includes human cell lines. Table 1 caption was therefore changed as “Doses of EDs normalized to the same unit of measurement, and their effects on human thyroid cell monolayers.”
Finally, the reviewer would like to have specification on the sex of the donors for cells lines and primary cells mentioned in both Tables. To this aim, a copy of original Tables 1 and 2 is included with specification of the donor’s sex, where known.
Reviewer 2 Report
The conclusion to the paper is that “SCREENED will move beyond the current state-of-the-art of in vitro assays evaluating effects of EDs on the thyroid function.” It is unclear what the SCREENED thyroid model represents from what is described in the paper. The SCREENED thyroid organoid culture model has never been published before making it difficult to rationally project what it will be useful for.
As mentioned in the paper “one of the key shortcomings of organoids is the lack of standardization” which underlines the need for standards in organoid cultures. The paper reviews briefly, if at all, results from proteomic and high-throughput RNA sequencing studies of human thyroid tissue. This could form the basis of a molecular standard for human thyroid tissue. A cellular standard might be the different types of cells in the organoid model vs human thyroid tissue. A histological or anatomic standard might be the structural architecture and physical dimensions of structures (follicle size) in the organoid model vs human thyroid tissue. A functional or physiologic standard might be how the organoid responds to say TSH exposure or other physiologic stimulus. Standards to define human thyroid tissue are important when one is considering how to effectively replicate it, as we do not have the option to do a rescue experiment in humans.
With proteomics and high throughput RNA studies there can be very minor changes noted across the multitude of targets and the question will be, is that a product of the culture system and how cells adapt and function in that system or a result that would have meaning for thyroid tissue in vivo. The verification of an organoid system to established thyroid tissue standards would elucidate the weaknesses or shortcomings of the three different 3D human thyroid organoid models proposed, as well as giving depth and support for any results that come out of experiments using these systems
This is an exciting area of research and I would encourage the authors to submit an article which takes a systematic approach to known thyroid tissue literature and expression databases to outline the architectural, physiologic, cellular, and molecular standards to which a 3D thyroid organoid culture should be held. The International Journal of Molecular Sciences does not have limits to data bases submitted, so it would be possible to present patterns of protein and RNA expression that are similar across different thyroid samples, rather than just naming the top three proteins as these are not likely to be the only targets of studies done on these organoid culture systems.
Author Response
Reply: We have provided a further clarification on the in vitro models that we would like to develop in the SCREENED project.
We have now added a new section and complemented the original section on proteomics and transcriptomics with a more thorough review of the current literature on the topic.
Round 2
Reviewer 1 Report
The authors have responded to all my remarks.